# **Urban Ozone Trends in Europe and the USA (2000-2021)**

Beth S. Nelson<sup>1,2,\*</sup> and Will S. Drysdale<sup>1,2,\*</sup>

<sup>1</sup>Wolfson Atmospheric Chemistry Laboratories, Department of Chemistry, University of York, Heslington, York, YO10 5DD, UK

Correspondence: Beth S. Nelson (beth.nelson@york.ac.uk) and Will S. Drysdale (will.drysdale@york.ac.uk)

#### Abstract.

Trends in urban maximum daily 8-hour average ozone concentrations, alongside annual ozone episodic and exposure metrics, across Europe and the United States of America were explored between 2000-2021. Using surface monitoring site data from the TOAR-II and European Environment Agency databases, piecewise quantile regression (PQR) analysis was performed on 353 time series (204 European, 149 USA). The PQR analysis permitted 2 break points over the period to balance the intent to describe changes over a large time period, while still capturing the abrupt changes that can occur in urban atmospheres. We found that there were many sites across Europe with high certainty increasing trends in the  $5^{th}$  and  $50^{th}$  quantiles, whereas the majority of high certainty trends in the  $95^{th}$  quantile were found to be decreasing. A similar pattern was observed across the USA, with  $5^{th}$  quantile trends increasing and  $95^{th}$  quantile trends decreasing, though a small but increasing number of sites showed a return to increasing trend at  $\tau=95$ . To group trends, hierarchical clustering with dynamic time warping was employed and these groups used to guide analysis. Clustering was typically regional across Europe and the USA, and increasing trends were identified across southern and the central alpine regions of Europe, and in California and the Intermountain West of the USA. Recent high certainty increasing trends in the Intermountain West may be related to warmer summers and increased wildfire events in the region, highlighting the need to monitor changing ozone trends with climate change, to assess human exposure risk to elevated levels of ozone.

#### 1 Introduction

Tropospheric ozone  $(O_3)$  is a greenhouse gas and an air pollutant harmful to human health and plant growth (Fleming et al., 2018; Mills et al., 2018; Szopa et al., 2021). It is a secondary air pollutant, formed from the photochemical reactions of primary pollutants  $NO_x$  ( $NO + NO_2$ ) and volatile organic compounds (VOCs). The chemistry of  $O_3$  formation is non-linear and the effect of changing precursor concentrations can cause  $O_3$  concentrations to increase or decrease depending on the photochemical regime found at a given location (Sillman et al., 1990; Sillman, 2002). Despite global successes in reducing primary pollutant emissions over the past few decades, global exposure to  $O_3$  has been increasing throughout the 21st century. This is particularly observed in urban areas, where the vast majority of the global population live, projected to increase to 68 % in 2050 from 55 % in 2018 (UN, 2019). In a study of 12946 cities located worldwide, the average mean weighted  $O_3$ 

<sup>&</sup>lt;sup>2</sup>National Centre for Atmospheric Science, University of York, York, UK

<sup>\*</sup>These authors contributed equally to this work.

concentration increased by 11 % between 2000 and 2019, and the number of cities exceeding the 2021 WHO peak season O<sub>3</sub> standard (60 µg m<sup>-3</sup>) increased from 89 % to 96 % (Malashock et al., 2022).

Due to the complexity of O<sub>3</sub> production, its trend direction, magnitude, and significance varies by location. The Tropospheric Ozone Assessment Report Phase I (TOAR-I) was an extensive review of trends in global surface ozone, covering 1970 - 2014 (Fleming et al., 2018; Gaudel et al., 2018). As part of the TOAR-I review, trends in two O<sub>3</sub> metrics were calculated globally over the period of 2000 - 2014: 4th highest daily maximum 8-hour O<sub>3</sub> in the warm season (4MDA8), and the number of days in a year with MDA8O<sub>3</sub> > 70 ppb O<sub>3</sub> (NDGT70). The study used data from 4801 global monitoring sites over this time period. For both of these metrics, downward trends were observed for most of the USA, and some sites in Europe. However, whilst trends were decreasing, over the period of 2010-2014 (2,600 sites utilised), some of these sites were located in regions that had the highest 4MDA8 and NDGT70 values, particularly California and southern Europe. These corresponded with the regions with the highest O<sub>3</sub> precursor emissions illustrating that downward trends had not yet reduced these peak metrics. Chang et al. (2017) further quantify summertime ozone trends in eastern North America and Europe, over the same period, using several metrics including: the monthly means of the daily maximum 8-hour average between April and September (DMA8) and NVGT070, a modified form of NDGT70 where only summertime days are counted. It is shown that trends in DMA8 across all sites are decreasing in areas in both regions, but in urban areas the decreasing trend is only seen in eastern North America (-0.25 ppb yr<sup>-1</sup>). For NVGT070 both eastern North America and Europe showed decreasing trends (-1.03 and -0.26 days yr<sup>-1</sup> respectively).

Since the 1990s, a general downward trend in urban  $O_3$  pollution has been observed in the United States (He et al., 2020). This reducing trend has been linked to stricter limiting regulations on the emissions of primary pollutants such as  $NO_x$  and VOCs. Although  $NO_x$  and VOC emissions in Europe have also been declining since the late 1980s, the trend in  $O_3$  is less clear due to large inner-annual variation, driven by climate variability and the dispersion and transport of pollutants from other regions (Jonson et al., 2006; Yan et al., 2018). Between 1995 and 2014, negative trends in the highest  $O_3$  levels across urban sites in Europe were identified due to pollutant emission restrictions across Europe. However, increasing background levels, particularly in northern and eastern Europe, make it difficult to identify strong trends in urban  $O_3$  when transboundary effects are considered (Yan et al., 2018). Despite this, a study of 93 suburban and urban sites across Europe identified notable enhancements in  $O_3$  seasonal and annual means between 1995 - 2012, even with the continuous downward trend in anthropogenic emissions across the continent (Yan et al., 2018).

In this study, we aim to investigate a longer time-series of 21st century trends (2000 - 2021). Here, we focus on the trends in urban O<sub>3</sub>, using monitoring site data from Europe and the United States of America across the 22 year period. We employ quantile regression and change point detection to construct trends that capture the broad structure of a complex time series, while remaining explainable with concise statistics. To gain a clearer understanding of the group behaviour of sites, we use hierarchical clustering with dynamic time warping, to group together time series with comparable structures within Europe and the United States of America independently. This is then used to investigate whether trends in MDA8O<sub>3</sub>, and both episodic and exposure-relevant O<sub>3</sub> metrics, are varying regionally. Our analysis focuses comparisons between warm season and cold season

MDA8O<sub>3</sub> values, as well as probing different changes across quantiles, to allow for an assessment of whether low or high O<sub>3</sub> distributions have different behaviours, across Europe and the USA, and between clusters.

# 2 Methodology

#### 2.1 Data Preparation

Hourly  $O_3$  data were obtained for urban sites in the USA and Europe using the TOAR-II (Schröder et al., 2021) and European Environment Agency (EEA) (European Environment Agency, a, b) databases. Those obtained from the EEA are categorised as urban traffic, urban background and urban industrial, while those from the TOAR-II database are categorised as urban and suburban. A full list of sites can be found in the supplementary information (table S1). TOAR-II data were retrieved between 2000-01-01 and 2021-12-31 (the latest available) and EEA data between 2000-01-01 and 2023-12-31, the latter being extended allowing break points to be fit around the changes due to the COVID-19 pandemic, though we remain focused on 2001-2021 in this study. Time series were quality controlled via threshold, persistence and variance checks similar to those by Wang et al. (2023):  $O_3$  concentrations were required to be between 0 and 500 ppbv, periods where an identical value was repeated 8 or more times in a row were removed, as were days where the difference between the minimum and maximum concentration was  $\leq 2$  ppbv. Data was subsequently aggregated as maximum daily 8-hour average  $O_3$  concentrations (MDA8 $O_3$ ), calculated by taking a rolling 8-hour mean, then selecting the highest value per day between 0700 - 2300 L. Six of the eight hours in a rolling window are required to be present for an MDA8 $O_3$  value to be valid.

After quality control, time series were required to have > 80 % data coverage between 2000-01-01 and 2021-12-31, and for the calculation of annual metrics each individual year was retained if it had more than 60 % coverage. Additionally, a small number of time series were removed following inspection due to large changes in the mean of data before and after a missing period, indicating possible sensor issues - these are listed in table S2. This resulted in 353 O<sub>3</sub> time series, 204 in Europe and 149 in USA.

#### 80 **2.1.1** Metrics

85

In addition to the series described above, we calculated six common health related  $O_3$  metrics which are explored in section 3.4 and used in conjunction with calculated trends in 3.5. More information on these metrics can be found in Lefohn et al. (2018) and Fleming et al. (2018).

- 4MDA8 The 4th highest MDA8O<sub>3</sub> value in the warm season (ppbv)
- NDGT70 Number of days in a year where MDA8O<sub>3</sub> > 70 ppbv (days)
- SOMO35 Subtract 35 ppb from the MDA8O<sub>3</sub> value and sum all positive values (ppbv days)
- 3MMDA1 Annual maximum value of 3 month rolling mean of maximum daily 1 hour O<sub>3</sub> (ppbv)

- 6MMDA1 As 3MMDA1 with a rolling 6 month mean (ppbv)
- AVGMDA8 Mean of the warm season MDA8O<sub>3</sub> (ppbv)

# 90 2.2 Trend Analysis

100

110

115

Trends were calculated on MDA8O<sub>3</sub> value using all data and subgroups of warm (April - September) and cold (October - March) season. It is common to remove the seasonal component of an O<sub>3</sub> time series to improve the accuracy of trends (Cooper et al., 2020) and here we subtract monthly mean climatologies from each time series.

These were calculated via quantile regression (QR) following the methodology in, and using code provided by, Chang et al. (2023). QR calculates a linear model that seeks to minimise the residuals with a defined proportion ( $\tau$ ) of the points above and below the fit line. For example, the scenario  $\tau = 0.5$  splits the data 50:50 above and below the line and can be considered analogous to a "median" trend line. QR has the advantage of being insensitive to outliers which is desirable for the longer-term trends being investigated here. The 1-sigma uncertainly for the QRs were calculated via a moving block bootstrapping method, where the data are subdivided into overlapping blocks that are  $n^{1/4}$  points wide (or ~20 points for 20 years of hourly data). These blocks are shuffled to generate a new time series, and replicated 1000 times, and the standard deviation of these replicates calculated. This uncertainty was subsequently used in the determination of the p-value, providing a metric for the significance of the trends.

Quantile regressions were calculated piecewise with 0 - 2 change points to allow the trends to capture large non-linearities. As urban concentration trends can change sharply (e.g. with policy intervention), this gives the model the freedom to represent these. Limiting the model to a maximum of two change points strikes a balance between capturing sufficiently large-scale changes while still being able to describe a ~20 year time series with a small number of coefficients.

To determine which change points to select for each time series, a range of candidate models were constructed, each with zero, one or two break points. These break points were restricted to be unable to occur within the first or last two years of the time series, nor could they occur within 5 years of each other. They were arbitrarily set to occur on the 1st January in a given year, which was deemed an appropriate degree of freedom given the 20 year span of the time series and locating the break points sub annually does not have a large effect on the resulting trends. For a time series that fully spanned the range of 2000-01-01 - 2021-12-31, 110 models would be created - one with zero break points, 18 with one break point, and 91 with two. All regressions were calculated at  $\tau = 0.05$ , 0.10, 0.25, 0.50, 0.75, 0.95 and 0.95.

To select 'change points' from the array of break points available, the model performance was evaluated via Akaike information criterion (AIC). AIC was chosen as the evaluation criteria owing to its penalty term penalising models with more break points that do not appreciably improve the model fit over the simpler case, The model with the minimum AIC was selected from the candidate pool as the best fit for a time series at a given  $\tau$ .

Trends have been collated into significance categories:  $p \le 0.05$  (high certainty), 0.05 (medium certainty), <math>0.10 (low certainty) and <math>p > 0.33 (very low certainty or no evidence) based off of the guidance of Chang et al. (2023). For the most part slopes where the p-value is > 0.33 are treated as 'no trend' regardless of their magnitude (generally we

observe that as the magnitude of the trend decreases so does its significance), though sometimes are given with a direction when required by a visualisation.

#### 2.3 Clustering

To aid analysis it was desirable to organise similar time series into groups, which was achieved by applying hierarchical clustering (HC) with dynamic time warping (DTW) used as a distance measure (Aghabozorgi et al., 2015). DTW provides a distance measure between two time series that allows for some deviation in the relation of features in time and one-to-many mapping of features, as opposed to use of e.g. the euclidean distance where mapping between series is one-to-one and cannot undergo deformation (Berndt and Clifford, 1994; Mueen and Keogh, 2016). We adopt a similar method to Reed et al. (2025) using the R package *dtwclust* (Sarda-Espinosa, 2024) for both DTW and HC steps, and used to calculate cluster validity indices (CVIs). Clustering was performed by region and time series type (e.g sites in Europe for the MDA8O<sub>3</sub> warm season). Time series were normalised and then, to provide clusters related to the range of *τ* values used in the QRs, aggregated monthly by calculating the *τ*<sup>th</sup> quantile per month. Metrics were not aggregated as they are all annual values. Clustering was calculated with 5 - 75 clusters, and then evaluated with 6 internal CVIs (Silhouette, Dunn, Calinski-Harabasz, COP, Davies-Bouldin and Modified Davies-Bouldin) where the latter 3 were inverted such that all can be maximised. The final selection for the number of clusters was determined as the mean of the optimal number of clusters suggested by each of the CVIs.

#### 3 Results and Discussion





As an initial overview, figure 1 shows the annual mean concentrations across the range of all the urban sites in this study. Additional daily mean aggregations are shown to produce day (0800 - 1900 L) and night (2000 - 0700 L) subgroups. European O<sub>3</sub> concentrations show a steady increase over the last 20 years, with the average across all data increasing from 21.2 ppbv in 2000 to 25.9 ppbv in 2019, the MDA8O<sub>3</sub> increasing from 30.8 to 35.3 ppbv and warm season MDA8O<sub>3</sub>. In the USA average O<sub>3</sub> has increased from 27.7 to 29.7 ppbv, but MDA8O<sub>3</sub> and warm season MDA8O<sub>3</sub> both decreased, from 41.5 to 40.9 and 49.9 to 45.1 ppbv respectively. Annual average urban NO<sub>2</sub> is also shown to provide some context to precursor concentrations - these sites were selected under the same criteria for site type and coverage as the O<sub>3</sub> data. Both Europe and the USA have seen reductions in NO<sub>2</sub>, 19.0 ppbv in 2000 to 13.0 ppbv in 2019 in Europe and 19.7 to 9.8 ppbv in the USA. The USA did see a NO<sub>2</sub> minima in 2020 of 9.3 ppbv, but increased to 11.8 ppbv in 2022, whereas Europe continued its decrease to 10.6 ppbv in 2022.

The other features that stand out appear in the sub groups containing the warm season, but are most clear in the warm MDA8O<sub>3</sub> series corresponding to years with significant heatwaves - 2003, 2006, 2015 and 2018 in Europe and 2012 in the USA. Heat waves are historically linked to enhanced O<sub>3</sub> concentrations, and are expected to increase in frequency (Schär et al., 2004; Russo et al., 2015; Shen et al., 2016; Otero et al., 2016; Gouldsbrough et al., 2022). The continued impact of heatwaves on high O<sub>3</sub> is however, dependant on the future levels of O<sub>3</sub> precursors and reductions and indeed this can be observed at some sites (Meehl et al., 2018; Otero et al., 2021; Li et al., 2025; Chang et al., 2025), but urban sites may take longer to reach levels where O<sub>3</sub> - temperature sensitivity is reduced, owing to a higher initial concentrations (Vazquez Santiago et al., 2024).

While QRs were calculated across a wider range of  $\tau$  values, this discussion focuses on  $\tau$  = 0.05, 0.5 and 0.95 as the omitted values did not sufficiently expand the results presented.

#### 3.1 Change Point Assignment

Regional differences were observed in the number (QR; 0 change points, PQR\_1; 1 change point, PQR\_2; 2 change points) of change point assigned to a time series. Figure 2 shows that MDA8O<sub>3</sub> series in Europe were rarely described by QR (1 - 3 % of series), whereas the United States of America was described by it 30 - 62 % of the time. As the European time series extends to 2023, some of these additional change points are likely due to the COVID-19 pandemic, and indeed examining the position of change points in time (figures S1 and S2) shows an increased number of change points in 2019 - 2021 in Europe, also reflected in the (fewer) change points seen in the USA. This also highlights that there are more change points focused at the beginning and end of the series generally which may indicate some edge effects from the selection process. However, we don't expect this to have impacted our results as the method demonstrates the ability to choose regressions with zero change points, we do not observe large swings in trends overall at these times and that there is a good amount of other change points assigned within the time period. Later comparisons use 2004 and 2018 as "snapshot" years informed by this, as to avoid these potential edge or COVID-19 effects and the major 2003 heatwave in Europe, whilst and still being near the beginning and end of the study period. In these comparisons, 2018 was used despite some European sites experiencing a less widespread heatwave, as it is

shown to have a lesser effect on the O<sub>3</sub> metrics (section 3.3), and figure S1 does not show a difference in change points from non-heatwave years. We do not observe any particular spatial pattern to the assignment of change points within regions (figures S3 and S4).

# 3.2 Trends in MDA8O<sub>3</sub> Across Quantiles





The number of urban sites with increasing and decreasing trends for the 5<sup>th</sup>, 50<sup>th</sup> and 95<sup>th</sup> quantiles, annually, as well as the degree of certainty of the trends in the MDA8O<sub>3</sub>, MDA8O<sub>3</sub> warm season and MDA8O<sub>3</sub> cold season data is shown in Figure 3. We use p-values to describe the certainty of each trend, as defined by Chang et al. (2023). Generally, there are a large proportion of high certainty positive trends in the 5<sup>th</sup> and 50<sup>th</sup> quantile in the Europe data set (range of 34 - 72 % of trends across the 20-year period). In contrast there are fewer trends of high certainty in the 95<sup>th</sup>, the majority of which are decreasing (between 5 - 25 % across the 20-year period). The warm season data shows a similar pattern, but with fewer high certainty increasing trends in the 5<sup>th</sup> and 95<sup>th</sup> quantile. Many more high certainty increasing trends are observed in the cold season only data, particularly in the 5<sup>th</sup> and 50<sup>th</sup> quantile data (range of 29 - 73 % across the 20-year period). The proportion of high certainty increasing trends in the cold season appears to be increasing with time.

In the USA MDA8O<sub>3</sub> trend data, the majority of 5<sup>th</sup> quantile trends are high certainty increasing trends (35 - 53 %) and the 95<sup>th</sup> quantile is dominated by high certainty negative trends (46 - 76 %), with a mixture found in the 50<sup>th</sup> quantile. This is consistent with the findings of Simon et al. (2015) who observed increasing O<sub>3</sub> trends across urban sites in the USA (1998 - 2013) at the lower end of the O<sub>3</sub> distribution, and decreasing trends at the upper end, leading to an observed compression of the O<sub>3</sub> range. In the warm season only data, there is a larger contribution of decreasing trends across all quantiles. This is particularly true in the 95<sup>th</sup> quantile data, where the vast majority of trends are high certainty decreasing trends (55 - 84 %). However, there is a notable change year-on-year in the USA 95<sup>th</sup> quantile warm season dataset that is worth some discussion. Although decreasing trends dominate, and the number of high certainty increasing trends is low (1 - 11 %), we observed a step-change. Between 2000 - 2008, 1 - 3 % of trends are high certainty increasing trends, however, after 2008, the number of sites showing a high certainty increasing trend increases rapidly from 2 to 16 by 2016, with the majority of these sites located in western and central US, and the top three largest magnitude slopes (medium-large), located in the south. In the cold season data, there is very little year-on-year change in the distribution of increasing and decreasing trends in the 5<sup>th</sup>, 50<sup>th</sup> and 95<sup>th</sup> quantile.

While certainty and magnitude tend to follow one another, to probe the trends more specifically we define a threshold of "medium magnitude" of 0.5 ppbv yr<sup>-1</sup>, and explore the moderate or higher certainty trends with and without this threshold. Figure S5 shows full distribution of trends in the groups discussed. A summary of the proportion of increasing and decreasing trends highlighting the proportion of moderate or higher certainty and medium or higher magnitude trends in 2004 and 2018 is presented in Figure 4.

Across the Europe data set at the start of the century (2004) 85 % and 78 % of slopes showed an increasing trend in the 5<sup>th</sup> and 50<sup>th</sup> quantiles respectively (73 % and 57 % of total at moderate certainty or higher respectively), compared to 74 % showing a decreasing trend in the 95 % quantile (31 % of total at moderate certainty or higher). By 2018, the number of

increasing slopes increased across the three quantiles, to 90 % and 86 % for the 5<sup>th</sup> and 50<sup>th</sup> quantile (74 % and 69 % of total at moderate certainty or higher), and from 26 % to 39 % in the 95<sup>th</sup> quantile (from 7 % to 15 % of total at moderate certainty or higher). This supports an increase in the number of Europe sites with increasing O<sub>3</sub> trends at the end of the two-decadal period compared to the beginning, for which the majority of sites showed increasing trends in the low-mid quantiles during both time periods. In the 95<sup>th</sup> quantile, the majority of sites are showing decreasing trends in both time periods, but a larger proportion of increasing trends is apparent in 2018 compared to 2004.





In comparison, 82 % and 49 % of slopes showed an increasing trend in the 5<sup>th</sup> and 50<sup>th</sup> quantiles of the USA data set at the start of the century (56 % and 36 % of total at moderate certainty or higher). Similarly to those in Europe, 95<sup>th</sup> quantile trends were decreasing the in USA (87 % - 68 % of total at moderate certainty or higher). However, in contrast to Europe, by 2018 fewer trends were increasing, down to 68 % in the 5<sup>th</sup> quantile (40 % of total at moderate certainty or higher), from 82 % in 2004 (56 % of total at moderate certainty or higher). In the 50<sup>th</sup> quantile, more trends are decreasing (51 % - 30 % of total at moderate certainty or higher) than increasing (49 % - 36 % of total at moderate certainty or higher), although more trends are increasing than decreasing when the data is filtered for moderate certainty or higher. In the 95<sup>th</sup> quantile, there was an increase in the number of sites with increasing MDA8O<sub>3</sub> in 2018 (26 %, 12 % of total at moderate or higher certainty) compared to 2004 (14 %, 3.9 % of total at moderate or higher certainty), and, whilst the majority of sites were still decreasing trends, a decrease in the number of these from 87 % (68 % of total at moderate or higher certainty) to 74 % (60 % of total at moderate or higher certainty) between 2004 and 2018 was observed. To summarise, between 2004 and 2018 a reduction in the number of increasing trends was observed at the 5<sup>th</sup> and 50<sup>th</sup> quantiles, but the high extremes (95<sup>th</sup> quantile) saw an increase in the number of increasing trends, coupled with a decrease in the number of decreasing trends.

Separating the MDA8O<sub>3</sub> trends into warm (April - September) and cold (October - March) seasons shows much clearer trends, and reveals a regionality in the direction of MDA8O<sub>3</sub> trends. Generally across Europe early century (2004) data, there more increasing trends in the cold season (72 %; 37 % of total at of moderate or higher certainty) compared to the warm season (25 %; 7 % of total at moderate or higher certainty) in the 95<sup>th</sup> quantile. Although there are a high number of increasing trends in the warm and cold season for the 5<sup>th</sup> quantile (88 % and 78 % respectively), there are many more increasing trends with a moderate or higher certainty and medium or greater magnitude in the warm season (15 %) compared to the cold season (4 %). This pattern is retained in the 2018 slope data, with an increase to 19 % of 5<sup>th</sup> quantile trends increasing at moderate or higher certainty and medium or higher magnitude in the warm season. A smaller increase in the cold season 5<sup>th</sup> quantile was observed, and a larger increase in the 50<sup>th</sup> quantile (from 7 % to 17 % of total at moderate or higher certainty and medium or higher magnitude). In summary, although we are generally seeing increasing trends across the quantiles in both 2004 and 2018, this is particularly prevalent for the lower extremes in the warmer months, and the 50<sup>th</sup> quantile cold season data, where the trends are not only increasing, but are of higher certainty and magnitude.

In the USA cold season at the start of the century, 92 %, 83 % and 45 % of trends are increasing in the  $5^{th}$ ,  $50^{th}$ , and  $95^{th}$  quantiles respectively, the majority of which are of moderate or high certainty in the  $5^{th}$  and  $50^{th}$  quantile. In comparison, there are fewer increasing trends in the warm season for the  $5^{th}$  (52 %),  $50^{th}$  (17 %) and  $95^{th}$  (9 %) quantiles. In the warm season, the majority of trends are decreasing in the  $95^{th}$  quantile (90 %), with 63 % of all trends in this quantile having a moderate or

higher degree of certainty, at a medium or larger magnitude. The majority of 50<sup>th</sup> quantile trends are also decreasing (82 %; 70 % at moderate or higher certainty), but with a smaller proportion with both moderate or higher certainty and medium or higher magnitude (14 %). Comparable trends are observed in the 2018 slopes, but with a small increase in the number of increasing trends of moderate certainty or higher in the warm season 50<sup>th</sup> (from 12 % to 16 %) and 95<sup>th</sup> (from 3 % to 10 %) quantile, and in the 95<sup>th</sup> quantile cold season (from 23 % to 32 %). In the cold season, the number of increasing trends with a moderate or high certainty and also medium or higher magnitudes in the 95<sup>th</sup> quantile has increased from 5 % to 14 % of the trends.

#### 3.3 Hierarchical Clustering of MDA8O<sub>3</sub> time series






So far, trends have only been separated spatially by region (USA or Europe), but it is clear when visualising these that there is more geographical structure. Figures 5 and 6 show this for the warm season MDA8O<sub>3</sub> trends (those for MDA8O<sub>3</sub> and cold season MDA8O<sub>3</sub> are shown in figures S6 - S9). From these, differences in the north and south of Europe, and the coasts versus central United States are observed. However, it is not trivial to chose sub-regions through inspection alone, so dynamic time warping coupled with hierarchical clustering (DTW, HC, described in section 2.3) was implemented to identify time series with similar features. Here, we discuss the clustering observed in the MDA8O<sub>3</sub> data at the 5<sup>th</sup>, 50<sup>th</sup> and 95<sup>th</sup> quantile. HC was performed for each of MDA8O<sub>3</sub>, MDA8O<sub>3</sub> warm season and MDA8O<sub>3</sub> cold season, separately for the United States of America and Europe.

The HC of the European data set indicates that most sites are well described by one cluster only, in the 5<sup>th</sup> and 50<sup>th</sup> quantiles across all three types of MDA8O<sub>3</sub> data aggregation (figure 7). However, a few additional clusters are introduced in the 5<sup>th</sup> and 50<sup>th</sup> quantile cold season data, as well as the 50<sup>th</sup> quantile all seasons and warm season data sets. In the cold season, these clusters don't appear to have any particular regionality in the 5<sup>th</sup> quantile. However, in the 50<sup>th</sup> quantile two new clusters are introduced, localised around south central and east Europe (clusters 3 and 4). In the 95<sup>th</sup> quantile (high extreme) data, more regional clustering is observed in the full MDA8O<sub>3</sub> data, and the warm season only data. This regional clustering is more clearly observed in the MDA8O<sub>3</sub> data, with clusters location in north eastern Europe (cluster 1); across the continent from south and central western France to east Europe (cluster 2); some selected sites from across this same area, but also including sites in Spain and the UK (cluster 3); northern France (cluster 4), the east most cluster (5), and north east (cluster 6). In the warm season, the majority of clustered sites are located across Europe, with some exceptions in the north east (cluster 2), north west (cluster 3), and Spain (cluster 4).

For all clusters in the 95<sup>th</sup> quantile MDA8O<sub>3</sub> European data, an enhancement in the mean AVGMDA8, 4MDA8, NDGT70, SOMO35, 3MMDA1 and 6MMDA1 metrics is observed in 2003, when an extensive European heatwave occurred (figures \$10 - \$15). Enhancement in other significant heatwave years are also visible in some data clusters in 2006 (clusters 1,2 and 5); 2015 (clusters 1 and 2); and 2018 (clusters 1, 2, and 6). This relates to the features seen in the (warm season) MDA8O<sub>3</sub> averages in figure 1, the 2003 heatwave is notably larger in the average presented there. Later enhancements may be less clear due to those heatwaves being less wide spread, or reductions in O<sub>3</sub> - temperature sensitivities in those locations. This is indicated by the heatwaves not being present in all clusters as the decades progress, but from this study it is not possible to say definitively whether one or both effects are causing this.

The HC of the United States of America data set has only one large cluster for the full MDA8O<sub>3</sub> data set, cold season, and warm season only data in the 5<sup>th</sup> quantile (figure 8). However, some clear regional clusters appear in the 50<sup>th</sup> and 95<sup>th</sup> quantile data sets across all three data types. The main secondary cluster sites are located in Florida, also including some sites along the Gulf of Mexico and the south in the cold season (cluster 2, or cluster 3 in the MDA8O<sub>3</sub> full 95<sup>th</sup> quantile data). Some clustering around the Intermountain West and the south west is observed in the full MDA8O<sub>3</sub> data set in the high extremes (cluster 2, 95<sup>th</sup> quantile). The east coast generally falls into the largest cluster which doesn't appear to have a strong regionality, except for the in full MDA8O<sub>3</sub> data 95<sup>th</sup> quantile, where there is an independent north east coast cluster (cluster 5). Also in this data type, and in the 95<sup>th</sup> quantile data, there is a cluster located in south central USA (cluster 1).

For the USA  $95^{th}$  quantile MDA8O<sub>3</sub> data, clusters behave much less uniformly than they do in Europe. Whilst all MDA8O<sub>3</sub> metrics are generally decreasing across all clusters (clusters 1, 3, 4, 5 and 6), many metrics are either clearly elevated, increasing across the 20-year period, or both, for cluster 7, located in and around Los Angles, in southern California (figures S16 - S21). This is also observed to a lesser extent in cluster 2, located more generally in the west and in the Intermountain West region of the United States. For cluster 7, increasing trends are observed in the full MDA8O<sub>3</sub> data set. However, the MDA8O<sub>3</sub> trends in cluster 7 are generally of low certainty (figures 6 and S7). In cluster 2, mean MDA8O<sub>3</sub> is elevated compared to other clusters, excluding cluster 7, and the slope in MDA8O<sub>3</sub> trends is generally quite small. However, these trends typically have a high degree of certainty (p < 0.05), which are typically negative in the first half of the vicennial, and positive in the second, observable in figures 6 and S7.

# 3.4 Spatio-temporal distribution of O<sub>3</sub> metrics, relevant to health and exposure




As the health/exposure O<sub>3</sub> metrics are annually aggregated, the DTW/HC method was not directly applied to these data sets, but observations from inspection of figures S10 - S21 are presented here. The NDGT70 and 4MDA8 metrics are generally used to investigate variations in the highest values of O<sub>3</sub>. High values of these metrics are typically an indicator of episodes of high photochemical O<sub>3</sub> production.

Across Europe, patterns in high 4MDA8 ( $\geq$  85 ppbv) and NDGT70 ( $\geq$  25 days) frequencies and site locations are similar (figures S10 and S11). In general, the only sites that consistently exceed these metrics are located in southern Europe. Exceptions to this can be seen in 2003, 2006 and 2018 for both metrics, known heatwave years, when a large number of high 4MDA8 and NDGT70 values occurs across the continent. However, the number of sites exceeding the aforementioned thresholds during these heatwave events generally reduces with time, with 89, 45, 27, and 15 exceedances in 4MDA8 and 68, 19, 6, and 8 exceedances in NDGT70 for 2003, 2006, 2015, and 2018 respectively.

For the USA metrics, a high number of sites across the country are found to have high 4MDA8 and NDGT70 values in the early years of the 20-year decade, from 2000 up until 2007 (figures S16 and S17). From 2008, there are few sites with exceedances, generally located in the west, and occasionally around the Intermountain West region of the US. Due to a lack of data availability in 2012, a reported heatwave year, it is not possible for us to comment here on whether this event led to elevated 4MDA8 and NDGT70 frequencies.

The SOMO35, AVGMDA8, 3MMDA1, and 6MMDA1 O<sub>3</sub> metrics are used to describe the general O<sub>3</sub> exposure, not just high O<sub>3</sub> events, important for assessing O<sub>3</sub> health burdens, rather than policy compliance. Across Europe, and across the 20-year period, values of SOMO35 are generally fairly low across the board (typically < 4000 ppb day) (figure S12). SOMO35 values of > 4000 ppb day are more widespread across Europe, excluding northern Europe, in 2003; and at a few sites across southern Europe over the years, typically in Spain, Greece, Switzerland and Italy. This is reflected in the AVGMDA8 metric, where the lowest values are typically found in the UK, Scandinavia, and north of mainland Europe (typically < 40 ppb) (figure S15). Slightly higher values of AVGMDA8 are observed in central and eastern Europe (41 - 45 ppb) in the early years (2000 - 2014). However, an uptick in AVGMDA8 is observed in eastern Europe in multiple years from 2015 onward, with many sites observing levels of 46-50 ppb, or even higher in 2018 (51 - 55 ppb). Few sites exceed 56 ppb, and those that do are located in southern Europe. The exception to this is once more in 2003, when AVGMDA8 were typically > 56 ppb across Europe, reaching 72 ppb in the alpine region of central Europe. Both the 3MMDA1 and 6MMDA1 metrics are low across Europe for the 20-year time period, with maximum values of 52 ppb and 44 ppb respectively observed in northern Italy in 2003 (figures S13 and S14). Despite lower values across the continent, the highest of these (31 - 40 ppb) is consistent observed in southern Europe and the central alpine region, consistent with previous studies (Wang et al., 2024).

For the USA data, these general O<sub>3</sub> exposure metrics are generally higher than the observed values for Europe. At the start of the 20-year period, the lowest values of SOMO35 were observed in the central and eastern USA (typically 1000 - 4000 ppb day), with higher values observed in the south (typically 4000 - 6000 ppb day), and the highest values found on the west coast (up to 11981 ppb day) (figure S18). This is consistent with previous studies, which have observed that southern California is a hotspot for O<sub>3</sub> pollution (Fleming et al., 2018; Wang et al., 2024). Over the 20-year period, SOMO35 levels in central, eastern, and southern USA generally reduce. However, values in the west and Intermountain West are found to increase or stay high, with SOMO35 values of 10973 ppb day observed in one site in 2020. Across the vicennial, all sites exceeding 7000 ppb day are located in the west, with a general decrease in the number of sites observing this limit over the 20 years. Similarly to the European data, the AVGMDA8 metric shows a similar pattern to the SOMO35 data, and by 2014 there is a clear divide between central, eastern and southern US, where the lowest values are observed (up to 50 ppb), and the west and south west, including the Intermountain West (51 - 80 ppb) (figures S21). This narrative is also reflected in the 3MMDA1 and 6MMDA1 metrics (figures S19 and S20).

### 3.5 Coupling MDA8O<sub>3</sub> trends with 6MMDA1 O<sub>3</sub> exposure metrics

To investigate how trends are developing in different clustered regions across Europe and the USA variations in the annual exposure metric (6MMDA1) are compared to the slopes and significance of trends in MDA8O<sub>3</sub> in 2004 and 2018. Here, we only look at sites that have been assigned a cluster (where the number of sites > 3), using the HC clusters analysis described in section 2.3 to explore whether regional clusters were showing similar trends. Figures 9 - 11 show the MDA8O<sub>3</sub> trend against 6MMDA1, coloured by the clusters shown in section 3.3. Figures S22 - S24 show these same scatters coloured by significance. In the European MDA8O<sub>3</sub> data and in the 5<sup>th</sup> quantile where sites are behaving more uniformly (one bulk cluster), we do not see a clear reduction in high certainty increasing slopes between 2004 and 2018, but we do see fewer high certainty decreasing

slopes in 2018. Across the cluster, which includes a broad range of sites across Europe, we observe similar maximum values of annual 6MMDA1 in the two years (c.a. 68 ppbv), but fewer 6MMDA1 values in the lower range in 2018 (20 - 40 ppbv). For the 50<sup>th</sup> quantile data, where there are more clusters, observe that a cluster located across southern mainland Europe (cluster 2), includes several sites with the higher 6MMDA1 values (> 60 ppbv), in both 2004 and 2018. However, the certainty and direction of the trends in MDA8O<sub>3</sub> is mixed. A smaller cluster of sites located in Spain observed lower values of 6MMDA1 in 2004 (20 - 40 ppbv), which increases to 40 - 50 ppbv by 2018. These sites also have high certainty increasing MDA8O<sub>3</sub> trends in both 2004 and 2018. More broadly across all clusters, we again observe fewer high certainty decreasing slopes in 2018 than 2004. In the 95<sup>th</sup> quantile, where the clearest regional clustering can be observed, generally higher 6MMDA1 values are seen in 2018 than 2004. There is no notable grouping of the clusters their 6MMDA1 values in either 2004 and 2018, but there is generally a broader spread of values in 2004 compared to 2018. The location of sites with high certainty increasing or decreasing values is also mixed, showing no clear regionality. When we compare 6MMDA1 values to the 95<sup>th</sup> quantile warm season only MDA8O<sub>3</sub> trends, we again observed a compression of the range of 6MMDA1 values, at the higher mixing ratio end (c.a. 25 - 70 ppbv in 2004, vs 40 - 70 ppbv in 2018). We also observe that clusters located in northern Europe have the smallest 6MMDA1 values in both years, and trends are generally increasing but with low certainty.

Average 6MMDA1 values across the USA are generally higher than those reported across Europe. Some clearer regional clustering can be observed in the 50<sup>th</sup> quantile MDA8O<sub>3</sub> USA data, with a cluster located in Florida (cluster 2) generally showing lower 6MMDA1 values (40 - 60 ppbv) in both 2004 and 2018. Additionally, these sites are generally high certainty decreasing trends, and this is reflected in both the cold season and warm season only MDA8O<sub>3</sub> trends. In contrast, sites located in the west show the highest 6MMDA1 mixing ratios in both 2004 and 2018 (c.a. 70 - 90 ppbv) (cluster 7), and these are typically high certainty increasing trends in both years. These sites also have the highest 6MMDA1 values in the 95<sup>th</sup> quantile (cluster 7), but increasing trends in this quantile are generally of low certainty. More interestingly, the 95<sup>th</sup> quantile cluster located in the west including the Intermountain West region of the USA (cluster 2), typically observes 6MMDA1 values of 55 - 70 ppbv in both 2004 and 2018, but there is a upward shift in the slope magnitude of this cluster from approximately -1 - 0.5 ppb per year in 2004, to -0.5 - 1.25 ppbv per year. In addition, most of these trends shift from being of low certainty in 2004, to high certainty increasing trends in 2018. These trends are also reflected in the warm season MDA8O<sub>3</sub> trends. Over the 21st century, the Intermountain West has experienced both increased wildfires, and hotter springs and summers, which can enhance photochemical activity forming O<sub>3</sub> from regional anthropogenic emissions (Lin et al., 2017; Li et al., 2021; Peterson et al., 2021; Iglesias et al., 2022). The higher magnitude and high certainty increasing slopes in MDA8O<sub>3</sub> may be linked to increases in one or both of these factors.

# 4 Conclusions






The goal of this study was to determine trends in O<sub>3</sub> metrics relevant to human exposure, in urban locations. Due to non-linear relationships with precursor concentrations, policy intervention or environmental changes may not have straightforward impacts on ambient O<sub>3</sub>. Trends in urban MDA8O<sub>3</sub> were calculated from de-seasoned monitoring site data across both Europe

and the USA. Piecewise quantile regression analysis was utilised to assess long-term trends. Piecewise trends were limited to two change points per time series to keep the focus on long-term trends and the largest changes. Generally more sites in the USA were described by a trend line with no change points. In comparison, sites in Europe were almost exclusively described by trends with 1 or 2.








By varying the selected quantile, quantile regression allows trends in different areas of the MDA8O<sub>3</sub> distribution to be quantified. The analysis presented here focuses on the  $5^{th}$ ,  $50^{th}$  and  $95^{th}$  quantiles, relating to low, medium and high values of MDA8O<sub>3</sub>. Additionally these trends were calculated for warm season and cold season subgroups. Across the 22 year period, there are many sites in Europe with high certainty increasing trends in the  $5^{th}$  and  $50^{th}$  quantiles, but the majority of medium-high certainty  $95^{th}$  quantile trends in Europe were found to be decreasing. Separating the trends by season shows that the majority of these decreasing trends are found in the warm season in all quantiles and very few trends were increasing in the  $95^{th}$  quantile with p < 0.33. In the USA, increasing trends are more frequently observed in the cold season, in all quantiles but to a lesser extent  $95^{th}$  quantile. Overall  $95^{th}$  quantile trends are dominated by sites decreasing, driven by the warm season trends, though a small number of these trends at higher quantiles do appear to be increasing in the Intermountain West.

Recent models show positive trends in HCHO to NO2 ratios over the USA and Europe demonstrate a tendency towards NO<sub>x</sub>-limited regimes (Fadnavis et al., 2025), but high heterogeneity within urban areas and differences between urban areas means that this behaviour cannot be generalised to all monitoring sites. Indeed, a primary challenge of interpreting monitoring station data is how to identify sites showing similar behaviour, without grouping based on pre-existing geographic regions. This is particularly true for urban sites, for which  $O_3$  precursor emissions are dependent on population density and growth, as well as climate and topography. Hierarchical clustering with dynamic time warping was shown to be effective in grouping similar time series and returned differing clusters when time series were aggregated at different quantiles which agreed with those observed qualitatively when mapping calculated trends. Clusters tended to be more homogeneous at lower values of  $\tau$ , with differing subregional behaviour becoming more apparent at the  $\tau = 0.95$ . In the high extreme trends, a cluster located in California was showing increasing trends across the 22 year period, but these were typically of lower certainty. Higher certainty increasing trends were observed in a cluster located in the Intermountain West in the latter half of the century, which were of lower magnitude. O<sub>3</sub> exposure metrics, SOMO35, AVGMDA8, 3MMDA1, and 6MMDA1 were typically higher in the USA than Europe, and the European sites that where AVGMDA8 > 56 ppbv were located in southern and the central alpine region of the continent. In the USA, sites located in the west observed SOMO35 > 7000 ppbv day more consistently over the 22 year period, though the number of sites exceeding this limit decreased in frequency with time. More interestingly, although 6MMDA1 values in the Intermountain West were similar in 2004 and 2018, the MDA8O3 trends shifted from low certainty in 2004 to high certainty increasing trends in 2018. This was also observed in the warm season trends, and may be linked to an observed increase in wildfire events in the area over the 22 year period, or warmer summers and springs as reported in previous studies (Lin et al., 2017; Li et al., 2021; Peterson et al., 2021; Iglesias et al., 2022). With the risk of increasing occurrences of warm summers, wildfires, and heatwaves, it is crucial that we continue to understand trends in MDA8O<sub>3</sub>, as this study has identified increasing trends in locations in the USA (the west and Intermountain West), and Europe (southern and central alpine region), that already experience hot and dry weather conditions, that may exacerbate human risk to  $O_3$  exposure.

Figure 1. Time series of annual average MDA8O $_3$ , O $_3$  and NO $_2$  for urban sites in Europe and the USA. Day and night subgroups correspond to the hours of 0800 L - 1900 L and 2000 - 0700 L respectively, warm and cold refer to the months April - September and October - March respectively.

**Figure 2.** Number of sites with no change point (QR, gold), one change point (PQR\_1, red) and two change points (PQR\_2, green) in the MDA8O<sub>3</sub>, MDA8O<sub>3</sub> cold season (October - March), and MDA8O<sub>3</sub> warm season (April - September) series, separated into Europe and United States of America regions.

Figure 3. Time series of trends in MDA8O<sub>3</sub> for annual (left), cold season (middle), and warm season (right), for Europe (top) and the USA (bottom). Within each region the panels are grouped by  $\tau = 0.05$  (top), 0.5 (middle), 0.95 (bottom).

Figure 4. Number of  $O_3$  time series across slope significance and slope magnitude categories. Those of medium significance (p > 0.10) and medium significance and medium magnitude slope are highlighted.

# MDA8O<sub>3</sub> Warm Season

**Figure 5.** Trends in warm season MDA8O<sub>3</sub> in Europe in 2004 and 2018. Trends are shown by an arrow where the angle between vertically up and horizontal corresponds to 2.5 - 0 ppbv / yr<sup>-1</sup>, and between horizontal and vertically down corresponds to 0 - -2.5 ppbv / yr<sup>-1</sup>. The few trends that have a magnitude greater than 2.5 ppbv have been clamped. Colour corresponds to the direction and significance of the trend.

# MDA8O<sub>3</sub> Warm Season

**Figure 6.** Trends in warm season MDA8O<sub>3</sub> in the United States of America in 2004 and 2018. Trends are shown by an arrow where the angle between vertically up and horizontal corresponds to 2.5 - 0 ppbv / yr<sup>-1</sup>, and between horizontal and vertically down corresponds to 0 - 2.5 ppbv / yr<sup>-1</sup>. The few trends that have a magnitude greater than 2.5 ppbv have been clamped. Colour corresponds to the direction and significance of the trend.

Figure 7. Clusters determined for MDA8O<sub>3</sub> (left), MDA8O<sub>3</sub> cold season (middle) and MDA8O<sub>3</sub> warm season (right) at  $\tau$  0.05 (top), 0.50 (middle) and 0.95 (bottom) in Europe. Clusters are coloured when there are > 3 sites within a cluster, otherwise these are assigned 'No Cluster'. Clusters are numbered starting with the cluster containing the most sites as 1 within each pane.

Figure 8. Clusters determined for MDA8O3 (left), MDA8O3 cold season (middle) and MDA8O3 warm season (right) at  $\tau$  0.05 (top), 0.50 (middle) and 0.95 (bottom) in the USA. Clusters are coloured when there are > 3 sites within a cluster, otherwise these are assigned 'No Cluster'. Clusters are numbered starting with the cluster containing the most sites as 1 within each pane.

Figure 9. For the years 2004 (left) and 2018 (right), scatter plots show that years' MDA8O<sub>3</sub> trend in (top to bottom) Europe  $\tau = 0.05, 0.5, 0.95$ , United States of America  $\tau = 0.05, 0.5, 0.95$  vs the 6MMDA1 for the same year. Colours correspond to clusters presented in figures 7 and 8.

**Figure 10.** For the years 2004 (left) and 2018 (right), scatter plots show that years' warm season MDA8O<sub>3</sub> trend in (top to bottom) Europe  $\tau$  = 0.05, 0.5, 0.95, United States of America  $\tau$  = 0.05, 0.5, 0.95 vs the 6MMDA1 for the same year. Colours correspond to clusters presented in figures 7 and 8.

Figure 11. For the years 2004 (left) and 2018 (right), scatter plots show that years' cold season MDA8O<sub>3</sub> trend in (top to bottom) Europe  $\tau$  = 0.05, 0.5, 0.95, United States of America  $\tau$  = 0.05, 0.5, 0.95 vs the 6MMDA1 for the same year. Colours correspond to clusters presented in figures 7 and 8.

*Code and data availability.* All data for this study can be downloaded from the relevant databases. This study used the r-packages *toarR* (Drysdale, 2024) and *saggetr* (Grange, 2019) to perform the downloads.

Code for performing the data download, analysis and production of figures for this manuscript can be found at https://doi.org/10.5281/zenodo.14538197 doi. (Drysdale and Nelson, 2025)

Author contributions. BSN and WSD equally contributed to all aspects of this manuscript's production

Competing interests. The authors declare that they have no conflict of interest.

Acknowledgements. The Viking cluster was used during this project, which is a high performance compute facility provided by the University of York. We are grateful for computational support from the University of York, IT Services and the Research IT team.

The Authors acknowledge Prof. James Lee for their scientific advice and Prof. David Carslaw and Dr Stuart Lacy for their advice on the statistical analysis. We also thank Dr Stuart Lacy for their help with SQL and very useful suggestions on managing large datasets.

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
