# Peer review of "Urban Ozone Trends in Europe and the USA (2000-2021)"

_EGUsphere, 2024_

## Author Comment (AC1)

We would like to thank both reviewers for their responses, and it was encouraging that both recognised the effort put into the first version of the manuscript. Both however, found that substantial reworking would be required for this to meet the threshold of publication in ACP. We have taken their advice, questions and criticisms onboard and taken the time to revise our method from the beginning. We believe that this has improved the manuscript and are thankful to ACP for allowing us the time to make these changes.

The major issue identified was two-fold – firstly that our use of median daily values was incompatible with the use of high or low quantiles in the regressions was not then probing truly more extreme values – and secondly the resulting numbers were less relevant for further interpretation and policy implication (the "why").

To address this, we began by recalculating trends for maximum 8-hour daily ozone values instead of daily medians. We also calculated trends for just warm season and cold season values. In addition to the trends, we calculated time series of 6 ozone metrics that are commonly used when interpreting ozone burden, helping to link our results with existing work. Finally, we expanded our methodology to include clustering of the time series. This adds a new angle to our interpretation of the trends, identifying similar sites not based on pre-defined geographic groupings, but by detecting those that are behaving similarly.

This has led to us refocusing the discussion of our new results and the major departure from the original manuscript is that the interpretation of NO2 trends alongside the O3 ones no longer features, as we prioritised O3 in the period of revision.

Below is our point-by-point response to both reviewers, with our responses in-line

-------------- RC1 --------------

This manuscript describes a trend analysis of urban ozone in Europe and USA over 2000-2021. I believe the authors have great ambition and spent a great deal of time putting the analysis together. Unfortunately, their approach and discussion appear to be premature and unskilled for ACP. In general, I will expect a vast and extensive rewritten if this manuscript is not rejected. Some major issues are pointed out as follows:

1. Methodology: Although using nonlinear methods such as Loess can visually identify the change points, are the authors really inspecting all the ozone and no2 time series (>500) and recording the change points for each individual location?

AR: The discussion of LOESS within our methodology did not help clarify our method as was originally intended. We wished to contrast the trends that LOESS could capture with the aim of summarising trends with values that represent more than a single point in time and that visual inspection of these was not viable. We have removed this from the discussion of our methodology which now focuses on our selection of piecewise quantile regressions using AIC

1. Methodology: AIC is most useful to avoid overfitting, so it can be used to determine if a model with change points (more parameters) is actually better than a model without change points (fewer parameters). But AIC does not tell us the optimal changepoint location, the authors should discuss how they select the change point locations. More importantly, how to select change points objectively (eg, Muggeo 2003; Chen et al., 2011), given that some change points may be hindered by data variability and not visually detectable. The authors should also properly define what they mean regarding

> change points. To me, the authors merely compare the trends between different periods, and see which locations have large trend differences. This does not really change point analysis in statistics.

Chen, C. W., Chan, J. S., Gerlach, R., & Hsieh, W. Y. (2011). A comparison of estimators for regression models with change points. Statistics and Computing, 21, 395-414.

Muggeo, V. M. (2003). Estimating regression models with unknown break-points. Statistics in medicine, 22(19), 3055-3071.

AR: We welcome the comment regarding the suitability of the AIC to identify the optimal change point model. As described in Section 2.2 circa L 108, candidate models were generated for each time-series comprising a mix of between 0-2 break points, with the break point locations varying according to the rules described in the text. The AIC was used to identify the optimal model from these candidates as quantified by the highest likelihood after accounting for the penalty term on the number of free parameters. While the AIC (or any metric on its own) does not provide the optimal changepoint location, the candidate model with a changepoint that best describes the data will have the highest likelihood and thus be preferred by the AIC.

1. Majority of US ozone studies show the ozone reductions since 2000 in response to emissions controls, but this study shows contradicted results. The fundamental problem of this study is that they use daily median ozone to conduct trend analysis, which is neither relevant to human health nor generally interesting. Medians can occur either daytime in one day or nighttime in another, mixing two makes the trends really ambiguous and difficult to interpret. I don't understand why the authors do not use MDA8 or daytime/nighttime observations, especially since this study is focused on urban ozone.

AR: Thank you for this suggestion. As discussed at the beginning of this response we have refocused on MDA8O3.

1. Scientific interpretations are essential for ACP. Most discussions have only scratched the surface and not provided sufficient interpretations. For example, NOx is not the only proxy for ozone production, disproportionate seasonal trends (winter increases and summer decreases), wildfires, and weather (2003 heat wave in Europe and 2012 heatwave in the eastern US) also play important roles (Cooper et al. 2012; Simon et al., 2015; Seltzer et al., 2020; Wells et al., 2021; Chang et al., 2023), but none of these factors are discussed.

Chang, K. L., Cooper, O. R., Rodriguez, G., Iraci, L. T., Yates, E. L., Johnson, M. S., ... & Tarasick, D. W. (2023). Diverging ozone trends above western North America: Boundary layer decreases versus free tropospheric increases. Journal of Geophysical Research: Atmospheres, 128(8), e2022JD038090.

Cooper, O. R., Gao, R. S., Tarasick, D., Leblanc, T., & Sweeney, C. (2012). Long-term ozone trends at rural ozone monitoring sites across the United States, 1990–2010. Journal of Geophysical Research: Atmospheres, 117(D22).

Seltzer, K. M., Shindell, D. T., Kasibhatla, P., & Malley, C. S. (2020). Magnitude, trends, and impacts of ambient long-term ozone exposure in the United States from 2000 to 2015. Atmospheric Chemistry and Physics, 20(3), 1757-1775.

Simon, H., Reff, A., Wells, B., Xing, J., & Frank, N. (2015). Ozone trends across the United States over a period of decreasing NOx and VOC emissions. Environmental science & technology, 49(1), 186-195.

Wells, B, Dolwick, P, Eder, B, Evangelista, M, Foley, K, Mannshardt, E, Misenis, C, Weishampel, A. 2021. Improved estimation of trends in US ozone concentrations adjusted for interannual variability in meteorological conditions. Atmospheric Environment 248: 118234. http://dx.doi.org/10.1016/j.atmosenv.2021.118234.

AR: We agree that more discussion of the impact extreme events such as heatwaves would be beneficial. A new section, using ozone metrics for the analysis, has been added. Whilst we have not included seasonal information (summer vs. winter), our study does explore the difference between the highest and lowest values in the data (5th, 50th, and 95th percentiles of the quantile regression). This has allowed us to comment on the changing trends of lower MDA8 values, typically expected during the winter months, and higher values (expected during the summer months, and extreme events).

1. If the authors aim to study extreme ozone events (Section 3.2), they should know the difference between the 95th percentile of daily medians and the 95th percentile of daily MDA8 or hourly observations, these are completely different concepts. Using daily medians to study extremes is completely misleading and unreasonable.

AR: Noted, this has factored into our methodology changes to MDA8O3 as discussed above.

1. All the figures with maps (eg Figs 3, 4, 9-12) have low quality.

AR: We were unsure from this comment on the exact issue with the maps, which were produced at high resolution. However, we have improved some visual aspects of these that we feel could have made them clearer.

l22 which WHO standards?

AR: 2021 guidelines 60 ug m-3 - text amended

l49 please clarify which analysis will end in 2021 or 2023.

AR: we focus on trends up to 2021, though as data were available in the database for Europe to 2023 we included this in our calculations. We have made this clearer in the methodology

l63 medians are not calculated through averaging.

AR: we now use MDA8O3

l64 please provide justifications. A visual inspection may be subjective.

AR: as part of the rework of the methodology we improved our QA, leading to less sites being removed via inspection. Only three are now removed from inspection alone due to issues, which, which are for "large changes in the mean of data before and after a missing period, indicating possible sensor issues"

l66 deseasonalized

AR: wording changed

l217 should the units of slope be ppb/decade or ppb/year?

AR: section removed

-------------- RC2 --------------

This study presents an analysis of the multi-year trends in surface ozone (O3) and nitrogen dioxide (NO2) mixing ratios at a few hundred urban and suburban locations across Europe and the USA that have near-complete time series between 2000 and 2021.

The predominant statistical approach the authors use is piecewise quantile regression (PQR) on deseasonalised daily median values of O3 or NO2 at each site. The authors permit the PQR to have two change points in the trend over the time period, subject to some restrictions such as change points having to occur on a 1st of January, there is at least 5 years between change points, and there are no change points in the first and last 2 years of the time period. Change points are different between each pollutant and each site. The author summarise, mainly visually, the extent to which the collection of sites show increasing or decreasing (or non-significant) trends in O3 and NO2 across the whole time series, how the trends change in magnitude and/or sign at change points, and in what years the change points occur. The summaries are presented separately for the sites in Europe and in the USA.

General comments

Substantial effort has clearly been expended in undertaking all the PQR (and other) statistical analyses and in thinking of ways in which to visually summarise the resulting datasets of trend directions, magnitudes and changes. These visual summaries are inventive and helpful for appreciating the distributions within these summary datasets.

However, the overarching issue with this work is "why"? Or, to reframe the "why" into two more specific questions: what is/are the scientific and/or policy questions motivating these particular analyses; and what do scientists and/or policy-makers learn from these analyses that isn't known before? With respect to the first of these: as the authors note in the Introduction, a lot of analyses of O3 and NO2 trends have come before; but the Introduction doesn't really identify what specific questions motivate this particular study, and why the authors' approach is well-suited to answer such questions. With respect to the second: the paper is essentially a description of the summary statistics of the analyses – there is very little detailed discussion of what the reader learns scientifically or policy-wise from these analyses.

A second issue requiring further justification is the use of daily median O3 and NO2 as the underlying measure of O3 and NO2 levels. The majority of metrics used to capture O3 levels are based on the daily maximum 8-hour mean. This latter way of defining a daily value for O3 level could as easily be calculated as the median daily value, so why was it not used? It also means that when this paper is referring to trends in high levels of O3 it is referring to trends in the highest daily median O3 levels which does not match the usual way of thinking about episodic high O3 levels in the literature or in air quality quantification.

AR: Thank you for this perspective, we have aimed to address this in the refactoring of the methodology to use MDA8O3 + policy relevant metrics, as well as the expansion to include clustering analysis as mentioned at the top of this response

What is the rational for choosing a 5-y period (2000-2004) as a time range over which to summarise trends in daily medians at the start of the full time-range of the datasets, but a 7-y period (2015-2021) as the time period over which to summarise trends in daily medians at the end of the full time-range of the datasets? Doesn't the length of time period used to quantify a trend potentially have some influence (bias) on the distribution of trend magnitudes and p-values? i.e., that this is not a like-for-like comparison?

AR: The original aim for this was to aid in summarising the trends – we weren't averaging trends in this period so impacts on magnitudes/p-values should not have been present. The periods were chosen to be as even as possible (5, 5, 5, 7) – but the inclusion of the 'odd' two years in the final group was arbitrary. We acknowledge that this was not clear, so when discussing differences between the beginning and end of the period, we have instead chosen 2004 and 2018 to simplify discussion. These are sufficiently near the beginning and end of the period as to be (potentially) different from one another, while being outside of the period where we see lots of change points being assigned (discussed in the updated methods)

What is the scientific merit in comparing average trend values across those sites with significant positive trends (and average trend values across those sites with significant negative trends) between different time periods and between different geographical areas, given that there are different numbers (and different identities) of sites contributing to each of those average trend values?

AR: The averaging of trends to provide an overview has been removed in favour of average concentrations / MDA8O3 + variety of subgroups (figure 1, section 3). Trends are reserved for the more detailed analysis as we agree it is not clear the merit in presenting averaged trends, especially with the subregional differences we highlight during the cluster analysis

At present there seems to be insufficient insight from the analyses to justify publication. A major revision would require substantial attention to the above-mentioned motivations and take-home messages, and attention to questions about use of median values.

 Minor and editorial comments

L4: The time series are described here as being 23 years long, but the date range given in the title and in the first line of the abstract comprises 22 years.

AR: This inconsistency has been corrected – text now correctly refers to the period of interest as 22 years, the only longer time series are the 24 year (2000-2023) ones originally acquired for the European data, but the focus is still on 2000-2021, which has also been clarified

L6: Need to specify is meant by "high" European O3 levels, i.e. what metric of O3 is used to define "high".

L8: In this sentence, is the word "trend" still referring to high O3 levels, or is it now referring to trends in some form of average O3 level?

L8: Is a 5-year period of 2000-2004 long enough to be confident of the direction and magnitude of a trend? Ozone concentrations are notoriously temporally variable.

L9: typo in "where"

AR: Abstract has been rewritten

L28-30: I don't understand the point being made in the last two sentences about citing locations with highest absolute values in 4MDA8 and NDGT70: all discussion to this point has been about trends not absolute values. The next paragraph returns to talking about trends again.

L31-L33: Again, I am struggling to understand the narrative here. In the first 3 sentences of this paragraph, it is noted how precursor emissions in in the USA and Europe have been declining since the late 80s/early 90s, but in a sentence at the end of the previous paragraph it talks about North America and Europe having highest precursor emissions. What is the point the authors want the reader to take away?

AR: Restructured these sentences for clarity

L49: There is an error in the time-series range quoted here ("2020-2023"). Neither the start or the finish year match those given in the paper's title, nor are the constituent numbers the same but accidentally typed in the wrong order.

AR: this has been fixed

L137: typo in "where"

AR: this has been fixed

The captions of Tables 2 and 3 are not clear enough. It needs to be made clear that the data in the table are numbers of sites having the trends specified. The captions imply that the data in the table are the trends.

AR: These tables have been removed in the restructure

Section 3.3: Needs to be citation to Figures 9 and 10 in this section.

Section 3.3: The values for the trend switches are given in ppb, but if these values describe trends shouldn't there by a temporal component to the unit?

AR: The original section 3.3 has been removed

Figure 13: Several sites don't appear to have increasing NO2 levels from 2020 onwards, which is what the caption and associated manuscript text description state that this figure shores.

AR: No longer present

---

## Author Response (AR2)

Many thanks to both reviewers for this round of comments. We have implemented these, the details of which are described below.

**Report #1**

I would like to thank the authors for taking an extensive revision of the manuscript. I would recommend a few technical revisions before potential acceptance.

1) This paper focuses on urban ozone, but the discussion of "urban" part is completely lost after introduction. It is desirable to add their implications specifically for urban chemistry and air quality.

We have added the following to the conclusions to more clearly frame the discussion w.r.t urban conditions

"The goal of this study was to determine trends in  $O_3$  metrics relevant to human exposure, in urban locations. Due to non-linear relationships with precursor concentrations, policy intervention or environmental changes may not have straightforward impacts on ambient  $O_3$ ."

"Recent models show positive trends in HCHO to NO2 ratios over the USA and Europe demonstrate a tendency towards NOx-limited regimes (Fadnavis et al., 2025), but high heterogeneity within urban areas and differences between urban areas means that this behaviour cannot be generalised to all monitoring sites."

"This is particularly true for urban sites, for which  $O_3$  precursor emissions are dependent on population density and growth, as well as climate and topography."

2) Although I know the rationale, it seems to lack an explicit explanation on why 2004 and 2018 are chosen as benchmark years (Fig 4 and further discussion).

Added the following to the end of section 3.1

"Later comparisons use 2004 and 2018 as "snapshot" years informed by this, as to avoid these potential edge or COVID-19 effects and the major 2003 heatwave in Europe and still being near the beginning and end of the study period. In these comparisons, 2018 was used despite some European sites experiencing a less widespread heatwave, as it is shown to have a lesser effect on the  $O_3$  metrics (section 3.3), and figure SI1 does not show a difference in change points from non-heatwave years."

**Minor comments:**

L28, technically the TOAR activity is not the first global review of trends in surface ozone, e.g. Cooper, O. R., Parrish, D. D., Ziemke, J., Balashov, N. V., Cupeiro, M., Galbally, I. E., ... & Zbinden, R. M. (2014). Global distribution and trends of tropospheric ozone: An observation-based review. Elementa, 2, 000029.

"... was the first global review of trends in surface ozone ..." replaced with: "... was an extensive review of trends in global surface ozone ..."

L32, these results are explicitly quantified in Chang et al. (2017), which is also a part of TOAR-I: Chang, K. L., Petropavlovskikh, I., Cooper, O. R., Schultz, M. G., & Wang, T. (2017). Regional trend analysis of surface ozone observations from monitoring networks in eastern North America, Europe and East Asia. Elem Sci Anth, 5, 50.

The following text has been added following L35:

"Chang et al. further quantify summertime ozone trends in eastern North America and Europe, over the same period, using several metrics including: the monthly mean of the daily maximum 8-hour average (DMA8) and NVGT070, a modified form of NDGT70 where only summertime days are counted. It is shown that trends in DMA8 across all sites are decreasing in areas in both regions, but in urban areas the decreasing trend is only seen in eastern North America (-0.25 ppb yr-1). For NVGT070 both eastern North America and Europe showed decreasing trends (-1.03 and -0.26 days yr-1 respectively)."

Section 2.1, some citations need to be fixed, e.g. (204 Europe, 149 USA).

These refer to the number of sites, the sentence has been adjusted for clarity: "This resulted in  $353 O_3$  time series, 204 in Europe and 149 in USA."

**Report #2**

The reviewers' reports on the original version of this paper acknowledged the effort expended in examining trends in the 22-year datasets of urban ozone but also raised substantive concerns regarding choice of ozone metric, the methodology of the trend analyses, and the overall motivation for the work.

The authors have taken these comments seriously and have essentially re-done all their trend analyses. The last paragraph of the Introduction is now much clearer about the aims of the work.

The authors now use the maximum daily 8-hour running mean (MDA8) rather than the daily median as their measure of daily ozone. They also investigate trends in other metrics that are commonly used to capture levels and human exposures to ozone. To compare how trends in ozone levels have changed (or not) through their 22-year time period they now more simply compare the magnitude of trend for a timepoint towards the end of their time period (2018) with that at a timepoint towards the start of their time period (2004). They also investigate trends separately for the warm season and for the cold season.

The revised paper also contains a new analysis, which is the grouping of time series with similar trends to visualise the extent to which trends at different sites do or do not group in distinct

geographic areas (within Europe and the USA separately). This is an interesting visualisation of the spatial homogeneity/heterogeneity in 22-year ozone trends across different measurement sites.

Trying to explain all the observed temporal and spatial patterns in ozone trends is of course a difficult matter, given the huge number of factors that influence ozone concentrations at any given time at any given location. The authors endeavour to provide some explanations, although discussion is still a bit lightweight in addressing the questions "so what have we learnt" and "what does it mean going forward"? The final sentence of the conclusion is one instance that does describe scientific conclusion. On the other hand, detailed interpretation would need very substantial investigations using full process-based atmospheric chemistry modelling, which is not what this paper is about, so perhaps the authors have gone as far as they can in providing 'user-relevant' information.

We appreciate the comments regarding the discussion, but agree that without a substantial increase to the scope of the work, we believe we have gone as far as we can. However, as part of our response to reviewer #1 comment 1), we have added briefly to the conclusions which may be relevant to the above.

The paper can be recommended for publication.

Some minor points:

The paper would have benefited from a careful proof read as there are quite a lot of missing or extra words.

Thank you for those you have already found below, in addition we have extensively checked the document and corrected spelling and grammatical errors we found.

The new Section 3.3 is potentially informative analysis but there is no cross-referencing to which of the many similar figures the reader needs to be looking at. Likewise, Section 3.4 contains a lot of description of different results of analyses but with no cross-referencing to tell the reader where they need to be looking for the visual support for these descriptions (just a blanket one-off statement that the observations are based on the twelve figures S10 to S21).

Thanks for bringing this to our attention, we have added more cross-refencing throughout sections 3.3 and 3.4 hopefully improving the readability of these sections.

L5: "the period"

Corrected

L30: Make clear that the 4MDA8 is calculated using only days in the warm season whilst NDGT70 is for all days in the year.

Updated to: "... 4th highest daily maximum 8-hour  $O_3$  in the warm season (4MDA8), and the number of days in a year with MDA8 $O_3$  > 70 ppb  $O_3$  (NDGT70) ..."

L65: Insert "to" before "be"

Done

L84: This definition of an MDA8 comes after the acronym MDA8 has already been used in the intro and methods sections.

Definition moved to 2.1

L88: The last part of this sentence doesn't make grammatical sense.

Language simplified to: "...and here we subtract monthly mean climatologies from each time series."

L89: The acronym QR appears here but has not yet been defined or described.

Now defined at the beginning of the paragraph: "These were calculated via quantile regression (QR) following the methodology in..."

L92: Delete extraneous "and the"

Done

L119: The start of this sentence is non-grammatical.

Reworded to: "To aid analysis it was desirable to organise similar time series into groups, which was achieved by applying hierarchical clustering..."

L142: "corresponding"

Done

L160: Reword to "closet to the beginning"

"...avoiding years closest to the beginning or the end of the series"

L197: Choose "were" or "are"

Done

L200: Sort out grammar in "Similar to the Europe"

"Similarly to those in Europe..."

L235: Suggest replacing "by region" with the more informative "by continent" and inserting the word "geographical" before "structure"

As this is the only location 'continent' would appear over 'region' we opted to reword to the following: "So far, trends have only been separated spatially by region (USA or Europe), but it is clear when visualising these that there is more geographical structure."

L239 onwards throughout the rest of Section 3.3: The reader's attention needs to be drawn to Figs. 7 and 8 and other supplementary figures, as relevant, when introducing the results from the hierarchical clustering and when referring to specific features shown in these figures.

As above, this section has had improved cross-referencing added.

L255: Visible where? Cite to the figure where these enhancements are visible.

Reference to figure 7 added earlier in paragraph

L361: Correct text to read "sites in Europe were almost exclusively"

Done

Figure 2 caption: The description of what metrics are plotted does not correctly reflect the text at the top of each column of plots says is plotted.

Text corrected to match figure

Figure 4: The font sizes need to be larger for all the labels, axis titles, panel titles, legend, etc.

The entire figure has been made larger, and we will liaise with typesetting to ensure legibility is preserved.